# Circulating AQP4 Levels in Patients with Cerebral Amyloid Angiopathy-Associated Intracerebral Hemorrhage

**DOI:** 10.3390/jcm10050989

**Published:** 2021-03-02

**Authors:** Paula Marazuela, Anna Bonaterra-Pastra, Júlia Faura, Anna Penalba, Jesús Pizarro, Olalla Pancorbo, David Rodríguez-Luna, Carla Vert, Alex Rovira, Francesc Pujadas, M. Mar Freijo, Silvia Tur, Maite Martínez-Zabaleta, Pere Cardona Portela, Rocío Vera, Lucia Lebrato-Hernández, Juan F. Arenillas, Soledad Pérez-Sánchez, Joan Montaner, Pilar Delgado, Mar Hernández-Guillamon

**Affiliations:** 1Neurovascular Research Laboratory, Vall d’Hebron Research Institute, Universitat Autònoma de Barcelona, 08035 Barcelona, Spain; paula.marazuela@vhir.org (P.M.); anna.bonaterra@vhir.org (A.B.-P.); julia.faura@vhir.org (J.F.); anna.penalba@vhir.org (A.P.); jesus.pizarro@vhir.org (J.P.); joan.montaner@vhir.org (J.M.); pilar.delgado@vhir.org (P.D.); 2Stroke Unit, Department of Neurology, Vall d’Hebron Hospital, 08035 Barcelona, Spain; olallaprvhir@gmail.com (O.P.); rodriguezluna@vhebron.net (D.R.-L.); 3Neuroradiology, Department of Radiology, Vall d’Hebron Hospital, 08035 Barcelona, Spain; carla_vert@hotmail.com (C.V.); alex.rovira.idi@gencat.cat (A.R.); 4Dementia Unit, Neurology Department, Vall d’Hebron Hospital, 08035 Barcelona, Spain; fpujadas@vhebron.net; 5Neurovascular Group, Biocruces Health Research Institute, 48903 Barakaldo, Spain; mariadelmar.freijoguerrero@osakidetza.net; 6Neurology, Son Espases University Hospital, 07120 Balearic Islands, Spain; silvia.tur@ssib.es; 7Department of Neurology, Donostia University Hospital, 20080 San Sebastián, Spain; mariateresa.martinezzabaleta@osakidetza.eus; 8Department of Neurology, Bellvitge University Hospital, L’Hospitalet de Llobregat, 08907 Barcelona, Spain; pcardonap@bellvitgehospital.cat; 9Stroke Unit, Department of Neurology, Ramon y Cajal University Hospital, 28034 Madrid, Spain; rovera78@hotmail.com; 10Stroke Unit, Virgen del Rocío University Hospital, 41013 Sevilla, Spain; lucia.lebrato.hdez@gmail.com; 11Stroke Program, Department of Neurology, Hospital Clínico Universitario, 47003 Valladolid, Spain; juanfarenillas@gmail.com; 12Clinical Neurosciences Research Group, Department of Medicine, University of Valladolid, 47003 Valladolid, Spain; 13Department of Neurology, Virgen Macarena University Hospital, 41009 Sevilla, Spain; soledad.perez.sanchez@gmail.com; 14Stroke Research Program, Institute of Biomedicine of Sevilla, IBiS, Virgen del Rocío University Hospital, University of Sevilla, 41009 Sevilla, Spain

**Keywords:** aquaporin 4, cerebral amyloid angiopathy, intracerebral hemorrhage, magnetic resonance imaging markers, functional outcome

## Abstract

Cerebral amyloid angiopathy (CAA) is a major cause of lobar intracerebral hemorrhage (ICH) in elderly patients. Growing evidence suggests a potential role of aquaporin 4 (AQP4) in amyloid-beta-associated diseases, including CAA pathology. Our aim was to investigate the circulating levels of AQP4 in a cohort of patients who had suffered a lobar ICH with a clinical diagnosis of CAA. AQP4 levels were analyzed in the serum of 60 CAA-related ICH patients and 19 non-stroke subjects by enzyme-linked immunosorbent assay (ELISA). The CAA–ICH cohort was divided according to the time point of the functional outcome evaluation: mid-term (12 ± 18.6 months) and long-term (38.5 ± 32.9 months) after the last ICH. Although no differences were found in AQP4 serum levels between cases and controls, lower levels were found in CAA patients presenting specific hemorrhagic features such as ≥2 lobar ICHs and ≥5 lobar microbleeds detected by magnetic resonance imaging (MRI). In addition, CAA-related ICH patients who presented a long-term good functional outcome had higher circulating AQP4 levels than subjects with a poor outcome or controls. Our data suggest that AQP4 could potentially predict a long-term functional outcome and may play a protective role after a lobar ICH.

## 1. Introduction

Cerebral amyloid angiopathy (CAA) is characterized by the deposition of amyloid in the walls of cerebral blood vessels [1]. The most common form of CAA is associated with the accumulation of amyloid-beta (Aβ) peptide in arterioles, capillaries, and leptomeningeal vessels and is frequently found in Alzheimer’s disease (AD) patients [2,3]. AD is the most common form of dementia worldwide, whereas CAA is the main cause of lobar intracerebral hemorrhage (ICH) [4,5]. ICH recurrence is one of the major complications of CAA, leading to substantial mortality and disability [6]. In addition to symptomatic ICH, transient focal neurological episodes and cognitive impairment independent of AD are also common clinical manifestations of CAA [7,8]. Despite the poor prognosis of CAA-related lobar ICH, no effective treatments are available, and a definitive diagnosis requires histopathologic demonstration by postmortem autopsy [9,10]. However, in clinical practice, the diagnosis of CAA is established following the modified Boston criteria based on clinical data and the presence of specific magnetic resonance imaging (MRI) markers [11]. Lobar cerebral microbleeds, cortical superficial siderosis, enlarged perivascular spaces in the centrum semiovale, and white matter hyperintensities are some of the frequent CAA radiological features detected by MRI [12,13,14,15].

Pathologically, Aβ peptides are generated by the sequential processing of amyloid precursor protein (APP) by β-secretase and γ-secretase, resulting in peptides consisting of 40 or 42 amino acids (Aβ40 or Aβ42, respectively). Aβ42 mainly accumulates in neuritic plaques in the brain parenchyma of AD patients, while Aβ40 is predominantly deposited on the walls of cerebral vessels, replacing smooth muscle cells and leading to vascular degeneration in CAA [16,17]. Although the cellular pathways explaining Aβ accumulation remain unclear, the most widely accepted theory is based on an imbalance between Aβ production and clearance. Several mechanisms have been described to remove Aβ from the brain, including proteolytic and microglial degradation, active transport across the blood–brain barrier (BBB), and perivascular and lymphatic drainage [17,18,19]. In fact, several studies have suggested that the perivascular drainage pathway may be impaired in CAA, which would lead to Aβ accumulation in vascular basement membranes [20,21,22].

Aquaporins are a family of water channel proteins that regulate water transport across cell membranes [23]. Aquaporin 4 (AQP4) is the most abundant water channel of the central nervous system and is mainly expressed by astrocytes and ependymal cells. AQP4 is highly localized to the perivascular astrocytic end-feet surrounding the glial limiting membrane of blood vessels [24,25]. Recent evidence suggests that AQP4 plays an essential role in the clearance of solutes between cerebrospinal fluid (CSF) and interstitial fluid (IF) through perivascular drainage, including Aβ clearance [26,27]. In this regard, several studies have described altered AQP4 expression and localization in AD and CAA patients [28,29,30]. However, most of these studies are based on postmortem analysis of human brain tissue, and the alteration of plasma AQP4 levels in AD or CAA patients is still unexplored. In the present study, we first aimed to study whether circulating AQP4 could be an indicative biomarker of CAA pathology and whether its determination would thus contribute to the diagnosis and prognosis of this disease. For this purpose, we analyzed the potential association of AQP4 levels with the main neuroimaging hallmarks of CAA in a multicenter cohort of patients with lobar ICH associated with CAA. We next determined the relationship between circulating AQP4 levels and other functional variables in this cohort.

## 2. Methods

### 2.1. Study Population

The study cohort consisted of 60 patients who presented with symptomatic ICH with a clinical diagnosis of CAA, as well as 19 age- and sex-matched control subjects. CAA–ICH patients had possible, probable or probable CAA with a supporting pathology diagnosis according to the modified Boston criteria [11]. Controls were healthy participants with no stroke history from the ISSYS (investigating silent strokes in hypertensives, a magnetic resonance imaging study) cohort who underwent brain MRI at a follow-up visit [31]. Controls were chosen for having no hemorrhagic events on their MRI scans. Sixty CAA–ICH patients were recruited in the neurology or stroke units from 10 different Spanish centers. CAA–ICH patients were >55 years old and had suffered at least one lobar ICH. Patients were excluded if they exhibited any deep intracerebral hemorrhage, presented microbleeds in the basal ganglia, internal or external capsule, thalamus or brainstem, or were being treated with anticoagulant therapy. The data obtained from the whole cohort included patient coding, inclusion date, demographic characteristics (age and sex), relevant vascular risk factors (hypertension, diabetes, and dyslipidemia), and brain imaging findings. The clinical and demographic data of both cohorts are shown in Table 1. Blood samples from the CAA–ICH cohort were obtained in a chronic state of the disease (13.6 ± 17.8 months after the last ICH) to avoid capturing the initial inflammatory process. Cognitive impairment was determined at the time of baseline visit (or blood draw) in all patients based on clinical history and neurological examinations. Additionally, functional outcome was assessed using the modified Rankin Scale (mRS) for neurologic disability. For a more detailed analysis, the CAA–ICH cohort was divided into two subcohorts based on the time point of the last functional outcome evaluation. In subcohort 1 (*n* = 35), mid-term outcome was assessed at 12 ± 18.6 months after the last ICH, the same time as when blood was drawn. In subcohort 2 (*n* = 25), blood was collected 15.8 ± 17 months after the last ICH, and long-term outcome was assessed at 34.4 ± 24.8 months after blood collection. A schematic representation describing the time points of recruitment and outcome evaluation for these cohorts is shown in Appendix A
Figure A1. Outcomes were dichotomized into good vs. poor: poor outcome was defined as mRS >3, and good outcome was defined as mRS ≤ 3.

The study was approved by the Clinical Investigation Ethics Committee of the Vall d’Hebron University Hospital, Barcelona, Spain (PR(AG)326/2014), and had the approval of the ethics committees of all of the participating centers. The study was conducted in accordance with the Declaration of Helsinki. All patients provided signed informed consent before inclusion.

### 2.2. MRI Protocol and Radiological Data

A brain MRI scan was obtained from all participants (1.5 ± 16.4 months after the last ICH). MRI examinations were acquired using a 1.5-T whole-body scanner system. The images obtained included axial T2-weighted turbo spin-echo, axial T1-weighted spin-echo, axial T2-weighted turbo fluid-attenuated inversion recovery (FLAIR), and axial T2*-weighted echo-planar gradient-echo sequences. All MRI scans were evaluated by the same neuroradiologist at Hospital Vall d’Hebron, who was blinded to the clinical and biological information.

The radiological characteristics of the CAA–ICH cohort obtained through MRI analysis are presented in Table 2. ICHs were defined as hypointense foci on the T2*-weighted images (diameter >5 mm), and their number and location were recorded. The presence, number, and distribution of cerebral microbleeds (CMBs; diameter <5 mm) were evaluated according to the Brain Observer Microbleed Scale [32]. White matter hyperintensities (WMHs) were defined as hyperintense signal lesions in T2-FLAIR or T2*-weighted images. Deep and periventricular WMHs were assessed according to the four-point Fazekas rating scale [33]. This scale scores damage from 0 to 4: (0) Absent or isolated foci of 3 mm, (1) foci of less than 5 mm (periventricular caps not included); (2) foci of more than 5 mm (periventricular caps not included); (3) beginning of confluence; (4) large confluent lesions (larger than 20 mm or two or more lesions merged). Severe WMH was defined as a score of 3 or 4. The WMH score was recorded in the hemisphere not affected by hemorrhage, except in cases when both hemispheres were involved. Perivascular or Virchow–Robin spaces are considered CSF-like signal lesions along the course of penetrating arteries. Enlarged perivascular spaces (EPVS) were counted in the basal ganglia and in the centrum semiovale (CSO) and were classified as moderate (≤20 EPVS) or severe (≥21 EPVS) according to the number found on axial T2-weighted MRI images [13,34]. Cortical superficial siderosis (cSS) was defined as the deposition of hemosiderin in the subpial layers of the cerebral cortex. The distribution and severity of cSS was categorized as focal (restricted to ≤3 sulci) or disseminated (>4 sulci) [11]. cSS contiguous to an ICH was not considered. Total cerebral small vessel disease (SVD) burden defined by Charidimou et al. was assessed using the principal MRI markers of CAA (lobar CMBs, WMH, EPVS, and cSS) [35]. The total SVD burden ranged from 0 to 6 points and was determined by counting the presence and grade of each of these 4 MRI features [35]. In our study, a high SVD burden was defined as a score ≥4.

### 2.3. Serum AQP4 Determination

Peripheral blood was collected in EDTA tubes, and serum was immediately separated by centrifugation at 1500 g for 15 min and stored at –80 °C. Only blood samples collected during a follow-up visit (at least 1.5 months after ICH) were considered for the analysis. The total AQP4 levels in serum were determined by enzyme-linked immunosorbent assay (ELISA) using the AQP4 Human Kit (Cusabio Biotech., Wuhan, China) following the manufacturer’s instructions. Optical density was measured at 450 nm in a Synergy™ Mx microplate reader (BioTek Instruments Inc., Vermont, USA). All samples were assayed in duplicate, and replicates with a coefficient of variation >20% were discarded from the statistical analyses.

### 2.4. Statistical Analyses

Statistical analyses were conducted with the SPSS 20.0 package (IBM Corporation, Armonk, NY, USA), and graphs were generated in GraphPad Prism 6 (GraphPad Software, La Jolla, CA, USA). Descriptive statistics were used to define the demographic data, clinical variables, and radiological characteristics of the CAA–ICH cohort. The normality of the continuous variables was assessed using the Kolmogorov–Smirnov test. For univariate analysis, the Mann–Whitney U-test and the Kruskal–Wallis test were used to evaluate the significant differences in the non-normally distributed variables between groups. Correlations were calculated as Spearman’s rho to compare two continuous variables, and the chi-squared test was used to assess the intergroup differences for the categorical variables. Binary logistic regression analysis was performed for ≥2 ICHs and cognitive impairment, including variables significantly associated with each end-point in the univariate analysis. A stepwise forward method was used in order to select those variables that fitted in the model. Data are expressed as the mean ± standard error of the mean or median values (interquartile range). A *p*-value <0.05 was considered statistically significant.

## 3. Results

### 3.1. Baseline Characteristics

The demographic, clinical, and radiological characteristics of the 60 CAA-related ICH patients and the 19 healthy control subjects included in the study are described in Table 1. There were no significant differences between the two groups in terms of age, sex, or APOE genotype. The control cohort was selected for not having ICH or detectable cerebral microbleeds by MRI. Some control subjects did show WMHs in both the periventricular and deep regions, but the prevalence was significantly higher in CAA-related ICH patients. AQP4 was detected in the serum from the control subjects and the CAA patients, although no significant differences were observed between groups.

The main MRI radiological characteristics of the CAA–ICH cohort are summarized in Table 2. All patients with lobar ICH fulfilled the Boston diagnostic criteria of probable or possible CAA. The majority of these patients (66.7%) had at least one lobar CMB, whereas no CMBs in the deep regions were detected. Overall, a high SVD burden score (63.3%), a high prevalence of severe WMH in the periventricular (68.3%) and deep regions (51.6%), a high degree of EPVS-CSO (36.7%), and disseminated cSS (35%) constituted the principal radiological features of the cohort.

### 3.2. AQP4 Levels According to Clinical and Radiological Characteristics

Circulating levels of AQP4 were evaluated in the CAA-related ICH cohort and analyzed for any correlations with the clinical and radiological characteristics. We determined that lower circulating levels of AQP4 were related to Apo**ε**4 and cognitive impairment in the univariate analysis (Table 3). However, only the presence of WMH in the periventricular areas remained independently associated with cognitive impairment after the adjustment by binary logistic regression analysis (Table A1 and Table 5). On the contrary, patients who had suffered a previous hemorrhagic stroke presented significantly lower levels of AQP4, whereas presenting a previous ischemic stroke did not result in changes in serum AQP4 levels (Table 3).

The neuroimaging analysis of the CAA–ICH cohort revealed a negative correlation between the number of lobar ICHs and AQP4 serum levels (Table 4). In fact, those patients with ≥2 lobar ICHs had significantly lower levels of AQP4 than those with only one lobar ICH (Figure 1A). After adjustment for significantly associated variables in the univariate analysis (Table 5), the logistic regression analysis confirmed that serum AQP4 levels, Apo**ε**2 allele, high SVD burden, and atrophy were independent predictors of ≥2 lobar ICHs (Table 5). In addition, a tendency (*p* = 0.052) toward lower AQP4 levels in patients presenting lobar CMBs was detected. When this variable was reclassified, serum samples from patients presenting with ≥5 CMBs showed lower AQP4 levels (Figure 1B). CAA–ICH patients presenting WMHs in deep regions also had significantly lower levels of AQP4 (Table 4). Remarkably, we did not detect an association between serum AQP4 concentration and the total MRI small vessel disease score validated for a pathological CAA presentation with symptomatic ICH [35].

### 3.3. AQP4 and Functional Outcome

We next studied the association between circulating AQP4 levels and functional outcome in two subcohorts divided according to the evaluation time point (Appendix A
Figure A1). A poor outcome was assessed in seven (20%) of the 35 patients from subcohort 1 and in eight (32%) of the 25 patients from subcohort 2. No statistically significant differences were found in the levels of AQP4 between subcohort 1 (1.87 (1.29–3.53) ng/mL) and subcohort 2 (2.38 (1.65–4.38) ng/mL) (*p* = 0.165). Interestingly, we found that AQP4 levels differed among CAA–ICH patients depending on the time when the outcome was assessed. First, in subcohort 1, when patient serum was obtained in parallel to the functional outcome evaluation, AQP4 levels did not differ between patients classified with good and poor outcomes (Figure 2A). In contrast, when the long-term functional outcome was assessed, those CAA–ICH subjects showing good outcomes presented significantly higher levels of AQP4 than subjects with a poor outcomes and controls (Figure 2B). In subcohort 2, AQP4 was the only variable associated with poor outcome in the univariate analysis (Appendix A
Table A2).

## 4. Discussion

In this study, we have shown for the first time the presence of AQP4 in the circulation in CAA-related ICH patients. Although no significant differences were found in circulating AQP4 levels between CAA–ICH patients and controls, an association between serum AQP4 levels and cerebral hemorrhagic load in the CAA cohort was detected. Furthermore, our data revealed that AQP4 could predict the long-term functional outcome in CAA patients after lobar ICH.

Interest in AQP4 research has grown considerably over the last few years. AQP4 is the most highly expressed aquaporin in the brain and is involved in the maintenance of the brain water balance under physiological and pathological conditions [24,29]. Several studies have demonstrated its implication in cerebral edema, ischemic stroke, traumatic brain injury, tumors, and neuromyelitis optica, among others [24,28,35]. Various studies have also found that AQP4 is necessary for the clearance of interstitial solutes, including Aβ, through the glymphatic system, suggesting a potential role of AQP4 in the physiopathology of AD and/or CAA [26,36].

To date, the vast majority of studies exploring the role of AQP4 in AD and CAA patients have described an altered distribution of this protein in diseased postmortem brain tissues. Wilcock et al. found reduced AQP4 expression in AD patients with moderate or severe CAA [37]. However, subsequent studies have demonstrated an increase in AQP4 immunoreactivity in AD and CAA brains compared with controls [38,39,40]. In particular, it was suggested that the AQP4 expression pattern could differ depending on the disease stage, with increased AQP4 immunodetection in CAA cases with the highest AD score [41]. In agreement with that, it has recently been described that the pattern of AQP4 immunodetection is different in gray matter compared to white matter and that it changes with age and with the severity of CAA [42]. Taking all of this into account, AQP4 expression in the brain may be associated with different factors, such as age, CAA severity, and specific neuroanatomical area. However, to the best of our knowledge, no studies have reported the modulation of AQP4 levels in the circulation in AD or CAA.

In this regard, we were able to detect AQP4 in serum from CAA-related ICH patients, although the levels did not differ from those in non-stroke control subjects. In agreement with previous studies [43,44], our results revealed that the presence of WMH in periventricular areas was also independently associated with cognitive impairment in the CAA–ICH cohort. In addition, we found lower circulating levels of AQP4 in patients with cognitive impairment in the univariate analysis. This result is aligned with findings in preclinical models, where behavioral tests revealed a cognitive deficit in AQP4 knockout mice [45,46,47]. Indeed, studies performed in AD and CAA transgenic mouse models have confirmed that AQP4 deletion promotes cognitive deficits and increases Aβ accumulation and synaptic damage, suggesting a possible contribution of AQP4 to Aβ clearance through the brain vasculature [48]. In contrast to these results, in a recent study conducted in 5xFAD mice, an accelerated model of AD with a lower vascular damage contribution, AQP4 deficiency did not induce alterations in Aβ accumulation or in memory deficit [49]. Nevertheless, experimental evidence suggests that AQP4 may be involved in the clearance of Aβ through perivascular drainage, thus playing a potential protective role in CAA pathology. However, in our study, we did not find an association between circulating AQP4 levels and the global SVD neuroimaging burden score, which has been associated with pathological CAA [35,50]. This result suggests that circulating AQP4 is not an indicative biomarker of CAA pathology. All the same, because AQP4 is related to solute clearance, further research should focus on studying AQP4 protein levels in CSF [26,27].

Remarkably, we found a clear association between serum AQP4 levels and CAA-related hemorrhagic lesions, including symptomatic ICH and CMBs, in the study cohort. Indeed, CAA–ICH patients presenting ≥2 ICHs and/or ≥5 lobar CMBs had lower levels of AQP4 than those patients without those hemorrhagic events, suggesting a potential protective role of this protein. Moreover, we demonstrated a negative correlation between the number of lobar ICHs and AQP4 serum levels, which is in agreement with other studies where AQP4 deletion worsened the neurological impairment and promoted neuronal death after ICH [51,52]. Indeed, AQP4 genomic variants were recently found to be independent predictors of the outcome after ICH in different populations [53,54]. Furthermore, we found that circulating AQP4 levels, together with the Apoε2 allele, atrophy, and high SVD burden variables, were independent predictors of presenting ≥2 ICHs. In this line, the Apoε2 allele is a well-known risk factor for ICH attributed to CAA, predisposing patients to recurrent bleeding [55,56,57]. Due to the specific localization of AQP4 at the astrocyte end-feet, it has been suggested that AQP4 could play a role in maintaining the BBB integrity [58]. Although most studies seem to point to a protective role for AQP4, it is still controversial whether AQP4 deletion alters the integrity of the BBB after ICH [59,60,61,62]. Further studies are needed to confirm the exact function of AQP4 in the brain following ICH, and our findings should be expanded to other cohorts presenting with ICH independently of a CAA etiology.

The present study also revealed that the analysis of serum AQP4 could potentially predict a long-term functional outcome in CAA–ICH patients. Our results showed that patients with a good long-term functional outcome presented higher circulating levels of AQP4 than patients with poor outcomes or healthy controls. In this area, we are aware of only one study that analyzed the levels of AQP4 in the circulation in neurological disorders, reporting higher levels of AQP4 in ischemic stroke patients after the acute phase of the disease [63]. Furthermore, in this same study, AQP4 was found to be an independent predictor of good neurological outcome in stroke patients, which is in agreement with our results [63]. Overall, the increase in circulating AQP4 levels in CAA–ICH patients presenting good long-term outcome and the decreased AQP4 levels in patients with a higher cerebral hemorrhagic load provide suggestive evidence of the potentially protective role of AQP4 after ICH.

Our study has some limitations. First, the sample size was relatively small, which could mask some associations not detectable after multiple testing corrections. Other limitations include the cross-sectional design of the study and the differences in the time points of the recruitment and functional outcome evaluation after the hemorrhagic episode. For this last reason, we decided to divide the total cohort into two subgroups, reducing the sample size even more. Taking all of this into consideration, further research should focus on larger patient cohorts with functional evaluation at different time points to explore whether our findings could be useful for the clinical prediction of long-term functional outcome in ICH patients. It would also be interesting to study the temporal profile of AQP4, as well as to evaluate the association between circulating AQP4 levels and the progression of neuroimaging markers with a follow-up blood draw and an MRI examination at different time points.

## 5. Conclusions

This study demonstrated, for the first time, that AQP4 can be detected in serum samples from CAA-related lobar ICH patients. Our data showed an association between circulating AQP4 levels and specific hemorrhagic neuroimaging features. In particular, we observed a decrease in AQP4 levels in patients presenting with more than two symptomatic lobar ICHs and in patients with more than five lobar CMBs detected by MRI. These findings suggest that AQP4 could play a protective role in CAA patients after ICH, potentially preserving BBB functionality, which could provide a possible therapeutic target for this pathology. Furthermore, because we found that circulating APQ4 levels were higher in those patients presenting a good long-term outcome, analysis of this serum biomarker might be considered an interesting candidate to be further investigated to improve the accuracy of the outcome prognosis in patients presenting with lobar ICH.

## Figures and Tables

**Figure 1 jcm-10-00989-f001:**
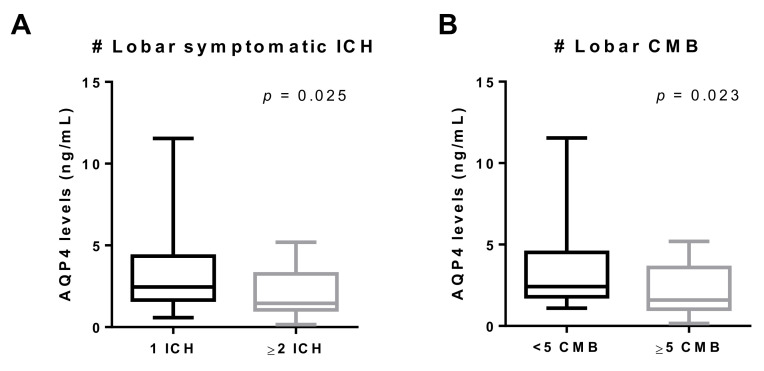
Association between AQP4 levels and hemorrhagic events in the CAA–ICH cohort. (**A**) Boxplot distribution according to the number of symptomatic ICHs (1 ICH, *n* = 43; ≥2 ICH, *n* = 17). (**B**) Boxplot distribution according to the number of lobar CMBs (<5 CMB, *n* = 34; ≥5 CMB, *n* = 26). ICH, intracerebral hemorrhage; CMB, cerebral microbleed.

**Figure 2 jcm-10-00989-f002:**
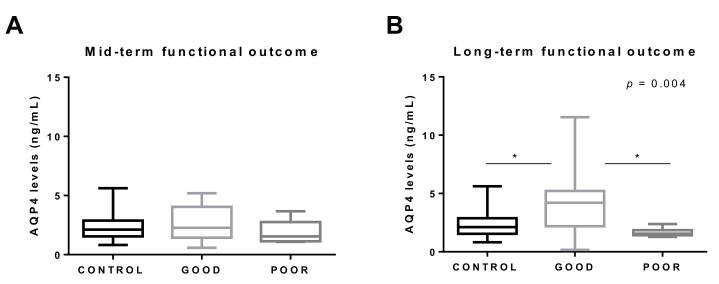
Association between AQP4 levels and functional neurological outcomes in the CAA–ICH cohort. (**A**) Boxplot distribution between good (*n* = 27) and poor (*n* = 7) mid-term functional outcomes (CAA–ICH subcohort 1) compared to controls (*n* = 19). (**B**) Boxplot distribution between good (*n* = 17) and poor (*n* = 8) long-term functional outcomes (CAA–ICH subcohort 2) compared to controls (*n* = 19). Outcomes were dichotomized into good vs. poor, and good outcomes were predefined as 0–3 on the modified Rankin Scale for Neurologic Disability. ICH, intracerebral hemorrhage; CMB, cerebral microbleed. * *p* < 0.05.

**Table 1 jcm-10-00989-t001:** Demographic, clinical, and radiological characteristics of the total cohort.

Variable	Control (*n* = 19)	CAA–ICH (*n* = 60)	*p*-Value
Age, years, median (IQR)	74 (73.5–74)	76.5 (71.5–70)	0.130
Sex, female, *n* (%)	10 (52.6%)	30 (50%)	1
Hypertension	19 (100%)	29 (48.3%)	**0.000**
Diabetes	6 (31.3%)	7 (11.7%)	0.063
Dyslipidemia	16 (78.9%)	17 (28.3%)	**0.000**
APOE genotype, ε2 carriers	1 (5.3%)	8 (13.3%)	0.679
APOE genotype, ε4 carriers	7 (36.8%)	14 (23.3%)	0.251
Lobar ICH	0 (0.0%)	60 (100%)	**0.000**
WMH, ***n*** (%)	9 (47.4%)	57 (95.0%)	**0.000**
CMB	0 (0.0%)	40 (66.7%)	**0.000**
Serum AQP4, ng/mL, median (IQR)	2.12 (1.63–2.67)	2.15 (1.44–4.12)	0.626

CAA, cerebral amyloid angiopathy; IQR, interquartile range; APOE, apolipoprotein E; WMH, white matter hyperintensity; ICH, intracerebral hemorrhage; CMB, cerebral microbleed; AQP4, aquaporin 4. *p*-Values below 0.05 are shown in bold.

**Table 2 jcm-10-00989-t002:** Radiological characteristics of the CAA–ICH cohort.

CAA–ICH (*n* = 60)	
Boston Criteria	
Possible	12 (20.0%)
Probable	45 (75.0%)
Probable with supporting pathology	3 (5.0%)
WMH	57 (95.0%)
Periventricular	51 (85.0%)
Moderate (1–2 Fazekas)	10 (16.7%)
Severe (3–4 Fazekas)	41 (68.3%)
Deep subcortical WMH	52 (86.7%)
Moderate (1–2 Fazekas)	21 (35.0%)
Severe (3–4 Fazekas)	31 (51.6%)
CMB	40 (66.7%)
Lobar CMB	40 (66.7%)
1–5	14 (23.3%)
6–10	9 (15.0%)
10–20	3 (5.0%)
>20	14 (23.3%)
Deep CMB	0 (0.0%)
Cerebellar CMB	4 (6.7%)
EPVS	53 (88.3%)
EPVS basal ganglia	52 (86.7%)
Moderate (1–20)	43 (71.7%)
Severe (21 to >40)	9 (15.0%)
EPVS CSO	41 (68.3%)
Moderate (1–20)	19 (31.7%)
Severe (21 to >40)	22 (36.7%)
cSS	30 (50.0%)
Focal	9 (15.0%)
Disseminated	21 (35.0%)
Atrophy	23 (38.3%)
Small vessel disease burden	
Low (0–3)	22 (36.7%)
High (4–6)	38 (63.3%)

Data are expressed as *n* (%).WMH, white matter hyperintensity; CMB, cerebral microbleed; EPVS, enlarged perivascular space; CSO, centrum semiovale; cSS, cortical superficial siderosis.

**Table 3 jcm-10-00989-t003:** Univariate analysis of AQP4 levels according to CAA–ICH demographic and clinical characteristics.

Variable	YES	NO	*p*-Value
Age	*r* = 0.147		0.264
Sex, female	2.19 (1.46**–**4.20) *n* = 30	2.08 (1.41**–**4.04) *n* = 30	0.684
Hypertension	1.85 (1.45**–**3.29) *n* = 29	2.89 (1.77**–**4.27) *n* = 28	0.102
Diabetes	2.60 (2.11**–**3.58) *n* = 7	2.14 (1.44**–**4.27) *n* = 48	0.435
Dyslipidemia	1.81 (1.41**–**2.89) *n* = 17	2.11 (1.44**–**4.41) *n* = 36	0.331
APOE genotype, ε2 carriers	1.75 (1.18**–**3.00) *n* = 8	2.30 (1.45**–**4.27) *n* = 52	0.317
APOE genotype, ε4 carriers	1.46 (1.03**–**2.60) *n* = 14	2.41 (1.65**–**4.20) *n* = 46	**0.028**
Cognitive impairment	1.69 (1.27**–**2.76) *n* = 30	3.09 (1.81**–**4.38) *n* = 30	**0.030**
Previous stroke	1.28 (0.99**–**1.67) *n* = 12	2.68 (1.69**–**4.35) *n* = 48	**0.002**
Previous ischemic stroke	1.53 (1.26**–**3.27) *n* = 4	2.68 (1.69**–**4.35) *n* = 48	0.261
Previous hemorrhagic stroke	1.12 (0.79**–**1.61) *n* = 8	2.68 (1.69**–**4.35) *n* = 48	**0.001**

Data are expressed as median nanograms per mililiter (interquartile range). *r*, Spearman’s rho correlation, CAA–ICH cohort; *n* = 60. *p*-Values below 0.05 are shown in bold.

**Table 4 jcm-10-00989-t004:** Univariate analysis of AQP4 levels according to CAA–ICH radiological characteristics.

Variable	YES	NO	*p*-Value
Interval between the last ICH and the date of blood collection	*r* = –0.052		0.703
Number of lobar ICHs	*r* = –0.307		**0.017**
WMH	2.04 (1.43–3.71) *n* = 57	4.46 (3.15–5.7) *n* = 3	0.163
Periventricular	2.02 (1.42–3.69) *n* = 51	3.29 (1.84–5.02) *n* = 9	0.092
Deep subcortical WMH	1.88 (1.42–3.67) *n* = 50	3.41 (2.04–5.19) *n* = 10	**0.045**
Lobar CMB	1.83 (1.41–3.79) *n* = 40	2.84 (1.93–4.25) *n* = 20	0.052
EPVS	2.26 (1.46–4.04) *n* = 53	1.81 (1.27–3.78) *n* = 7	0.718
EPVS basal ganglia	2.30 (1.47–4.12) *n* = 52	1.63 (1.27–3.78) *n* = 8	0.521
EPVS CSO	2.45 (1.50–4.36) *n* = 41	1.84 (1.10–3.00) *n* = 19	0.144
cSS	2.11 (1.16–4.38) *n* = 30	2.15 (1.65–2.90) *n* = 30	0.988
Chronic Infarct	1.53 (1.27–2.02) *n* = 13	2.52 (1.55–4.19) *n* = 44	0.146
Atrophy	2.26 (1.43–4.00) *n* = 23	2.04 (1.50–4.04) *n* = 37	0.715
High SVD burden (score 4–6)	2.35 (1.27–4.34) *n* = 23	2.03 (1.81–2.90) *n* = 37	0.570

Data are expressed as median nanograms per mililiter (interquartile range). *r*, Spearman’s rho correlation. CAA–ICH cohort: *n* = 60. ICH, intracerebral hemorrhage; WMH, white matter hyperintensity; CMB, cerebral microbleed; EPVS, enlarged perivascular space; CSO, centrum semiovale; cSS, cortical superficial siderosis; SVD, small vessel disease burden. *p*-Values below 0.05 are shown in bold.

**Table 5 jcm-10-00989-t005:** Binary logistic regression for cognitive impairment and ≥2 ICHs.

	Regression Cognitive Impairment	Regression ≥ 2 ICHs
Variable	OR (95% CI) *p*-value	OR (95% CI) *p*-value
AQP4	-	0.520 (0.286–0.976) ***p* = 0.042**
APOE genotype, ε2 carriers	-	22.536 (1.989–255.296) ***p* = 0.034**
WMH periventricular	10.545 (1.227–90.662) *p* = 0.032	-
Atrophy	-	6.167 (1.080–35.213) ***p* = 0.004**
High SVD burden (score 4–6)	-	11.280 (1.109–114.739) ***p* = 0.025**

Binary logistic regression analysis was performed with variables associated with cognitive impairment and/or ≥2 ICHs in univariate analysis. The results are given as odds ratios (ORs) with 95% confidence intervals (CIs) and *p*-values. SVD, small vessel disease. *p*-Values below 0.05 are shown in bold.

## Data Availability

The data that support this study are available on request from the corresponding autor, M.H.-G.

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
