# Peer review of "Circulating AQP4 Levels in Patients with Cerebral Amyloid Angiopathy-Associated Intracerebral Hemorrhage"

_jcm, 2021, doi:10.3390/jcm10050989_

Round 1
Reviewer 1 Report
The authors analyzed AQP4 levels in the serum of CAA related ICH pts and 19 non stroke subjects. The patient cohort was divided according to the time of last follow up. No differences in AQP4 levels were observed in patients and controls. However AQP4 levels were lower in pts with >2 lobar ICH and >5 lobar microbleeds and low AQP4 levels seem to correlate with a worse outcome
The paper is interesting since APQ has been observed to be involved in the clearance of solutes of CSF. However I have some major and minor concerns. Introducion: the final aims and the clinical translation/utility of the study should be clarified.
Methods: Why did the authors selected ICH CAA related population rather than CAA patients? Authors should explain this point. Controls are healthy participants (how were they selected? from general population? of neurological non stroke pts?) Since it is a prospective study authors should have defined recruitment time (time/range time from ICH) as well as time for serum sample obtainment for APQ4 dosage. Instead it appears as authors adapted the timetable to the study results. Also the strategy of grouping patients in two cohorts is unclear. I would consider to analyze patients in a unique cohort. MRI protocol and radiological data should be joined in a unique paragraph.
Did the authors assessed APQ4 only in serum or did they analyze samples also of CSF?
Results: Table 1 and 2 share some repetitions and should be joined in a unique table. Table 3 should be moved in supplementary data whereas the results of multivariate analysis should be reported in the paper. A temporal profile of APQ4 is not performed.
Discussion: the sample size is small and the results ( pts with >2 ICh or more than 5 microbleeds have lower levels of AQP4 ) are a bit arbitrary. What is the criteria to stratify patient severity on the base of number of ICH and microbleeds?
Author Response
The authors analyzed AQP4 levels in the serum of CAA related ICH pts and 19 non stroke subjects. The patient cohort was divided according to the time of last follow up. No differences in AQP4 levels were observed in patients and controls. However AQP4 levels were lower in pts with >2 lobar ICH and >5 lobar microbleeds and low AQP4 levels seem to correlate with a worse outcome.
We want to thank the reviewer for the time taken to review our manuscript and for the accurate interpretation of the results. We hope that with the new changes and corrections, the reviewer and the editor will find our manuscript more appealing and suitable for the publication in the Journal of Clinical Medicine. According to the reviewer’s comment, the manuscript has been carefully revised by MDPI English Editing Services to improve the language and format of the document.
- The paper is interesting since APQ has been observed to be involved in the clearance of solutes of CSF. However I have some major and minor concerns. Introduction: the final aims and the clinical translation/utility of the study should be clarified.
We thank the reviewer for this comment and highlight the interest of our manuscript. Following the reviewer’s suggestion, introduction and conclusion have been modified in the new version of the manuscript in order to clarify the objectives and the clinical translation of the study (page 2; introduction section & page 10; conclusion section)
- Methods: Why did the authors selected ICH CAA related population rather than CAA patients? Authors should explain this point.
The definite CAA diagnosis requires a full post-mortem examination or histopathological confirmation of the evacuated hematoma or brain biopsy. Thus, tissue samples from living patients are usually unavailable. However, in the clinical practice, the non-invasive diagnosis of CAA is established following the modified Boston criteria based on clinical and radiological data [1,2], as specified in the introduction (page 2, highlighted in yellow). Therefore, to delimit a clear CAA phenotype, only patients with a possible or probable CAA diagnosis, which involves the presence of at least one lobar ICH (CAA-related ICH cohort) and not presenting microbleeds in deep brain regions in the MRI were selected in the study, as defined by the Boston clinical diagnosis criteria.
- Controls are healthy participants (how were they selected? from general population? of neurological non stroke pts?)
Control cases were recruited in a previous study and were part of the ISSYS longitudinal cohort (Investigating Silent Strokes in hYpertensives, a magnetic resonance imaging study), as specified in the study population section (page 3, methods section, highlighted in yellow). Because one of the primary aims of our study was to analyze the association between circulating AQP4 levels and the main CAA radiological features, one of the requirements for the study design was to have access to MRI information also in controls. The objective was to discard possible CAA pathology in those cases. In this regard, all control subjects were free of dementia and stroke at the baseline visit and were selected for not presenting hemorrhagic lesions in the MRI as specified in page 3 (methods section, highlighted in yellow) and in page 4 (result section, highlighted in yellow).
- Since it is a prospective study authors should have defined recruitment time (time/range time from ICH) as well as time for serum sample obtainment for APQ4 dosage. Instead it appears as authors adapted the timetable to the study results.
Blood samples from the total CAA-ICH cohort were collected in a follow-up visit at 13.6 ± 17.8 (mean ±SD) months after the last ICH as specified in page 3 (methods section, study population, highlighted in yellow). We understand and recognize that the variability in the time range from the ICH to recruitment is an important limitation in the study protocol, as discussed in page 10 (highlighted in yellow). However, it is important to note that no correlation was found between AQP4 levels and the interval between the last ICH and the date of blood collection, as specified in the first row of Table 4.
We would like to mention here that the primary outcome of our research line was to find out specific biomarkers of CAA and to detect potential association of these markers with the main radiological hallmarks of the disease. The discovery phase was planned in a chronic stage of the disease, at least 2 months after the last ICH, to avoid confounding factors due to the initial inflammatory process, but without a limit time after the last ICH event. Because the progression of CAA may be understood as a chronic pathological process, we considered that the recruitment time of the ICH-CAA cohort was not a limiting factor to achieve the primary objective.
- Also the strategy of grouping patients in two cohorts is unclear. I would consider to analyze patients in a unique cohort.
The reviewer points a very relevant remark. The criteria for dividing patients into two cohorts was the time of functional outcome’s last evaluation. As it is shown in Supplementary Figure 1 (page 11), in subcohort 1, the functional evaluation was undertaken at the follow-up visit, when blood was drawn. However, in the other part of the cohort (subcohort 2), the modified Rankin was evaluated much later than the blood collection. In this regard, the information that might be obtained from blood samples of each subcohort is not comparable, especially in terms of potential utility for long-term functional prediction. As a summary, in subcohort 1, we cannot consider AQP4 levels in blood as a predictor of outcome because Rankin was evaluated at the same time as when blood was collected. In subcohort 2, as blood samples were collected 34.4±24.8 months before Rankin’s evaluation, we can consider studying whether blood AQP4 levels could be a long-term prognostic biomarker. We have attempted to clarify it by modifying Supplemental Figure 1 (page 11).
For these reasons, and although we agree with the reviewer that the ideal situation would have been to obtain functional outcomes at the same time for the whole cohort, the characteristics of our recruitment protocol do not allow us to analyze the AQP4 levels in a unique cohort according to their functional outcome. We recognize that it is an important limitation in the study, as we already pointed out in page 10 (discussion section, highlighted in yellow).
- MRI protocol and radiological data should be joined in a unique paragraph.
Following the reviewer’s suggestion, MRI protocol and radiological data have been joined in a unique paragraph in page 3.
- Did the authors assessed APQ4 only in serum or did they analyze samples also of CSF?
We thank the reviewer for this question. We agree that CSF analysis would certainly be of interest. Unfortunately, in the initial study protocol, no CSF samples were considered. The initial design was based on an observational, prospective and multicenter study to generate a cohort of patients with lobar ICH clinically diagnosed as probable CAA by MRI. The inclusion criteria included obtaining blood samples at least 2 months after the last ICH (to avoid the initial inflammatory process), as well as performing a brain MRI scan on each patient, among other criteria mentioned in page 3 of the manuscript (methods section, highlighted in yellow). However, CSF collection was not considered. In this line, we have suggested it as an interesting consideration for future research (page 9).
- Results: Table 1 and 2 share some repetitions and should be joined in a unique table. Table 3 should be moved in supplementary data whereas the results of multivariate analysis should be reported in the paper. A temporal profile of APQ4 is not performed.
Following the reviewer’s suggestion, Table 1 has been modified in the new version of the manuscript to avoid repetitions with Table 2. The classification of WMH and CMBs was eliminated from Table 1 since the information was detailed in Table 2. However, we consider that both tables are necessary since Table 1 refers to the differences between cases and controls, while Table 2 refers to the radiological features of the CAA-ICH cohort. Additionally, Supplementary Table 2 has been moved to the main document (Table 5 in the new version of the manuscript), as recommended by the reviewer. We consider that Table 3 should be in the main document as well. In fact, Table 3, together with Table 4, show those variables that were originally associated with AQP4 levels and were subsequently used for the logistic regression analysis (Table 5 in the new version of the manuscript).
On the other hand, as the reviewer states, a temporal profile of AQP4 level determination was not performed. We appreciate the reviewer’s suggestion and agree that it would be useful to study the temporal profile of circulating AQP4 after an ICH. However, in the initial study protocol, serial serum sampling was not planned at follow-up visits. Therefore, the only blood sample available is the one corresponding to the baseline visit after the ICH. Although we already mentioned this point in the discussion of the manuscript, we have tried to emphasize this limitation in the new version of the manuscript (page 10).
- Discussion: the sample size is small and the results (pts with >2 ICh or more than 5 microbleeds have lower levels of AQP4) are a bit arbitrary. What is the criteria to stratify patient severity on the base of number of ICH and microbleeds?
We agree with the reviewer that the sample size is small, as we discussed in the limitations of the study (page 10, discussion section, highlighted in yellow). We would like to remark that this study has been part of a multicenter effort to obtain a reasonable sample size to achieve our objective, considering the restrictive inclusion criteria to ensure a clear diagnosis of CAA.
On the other hand, ICH recurrence is one of the main complications of this pathology, reaching 10% per year after a first lobar ICH event and [3] leading to high rates of mortality and disability [4]. For this reason, we decided to stratify the hemorrhagic events in our cohort into one lobar ICH or ≥2 lobar ICH (which implies ICH recurrence). Furthermore, the presence and number of CMBs were evaluated according to current criteria [5]. Experts in the field have previously used the criteria of presenting more than 5 CMBs as a phenotype of multiple microbleeding [6–8]. Moreover, the presence of ≥5 CMBs has been previously associated with a high risk of hemorrhagic recurrence [9].
References:
- Linn, J.; Halpin, A.; Demaerel, P.; Ruhland, J.; Giese, A.D.; Dichgans, M.; Van Buchem, M.A.; Bruckmann, H.; Greenberg, S.M. Prevalence of superficial siderosis in patients with cerebral amyloid angiopathy. Neurology 2010, 74, 1346–1350, doi:10.1212/WNL.0b013e3181dad605.
- Knudsen, K.A.; Rosand, J.; Karluk, D.; Greenberg, S.M. Clinical diagnosis of cerebral amyloid angiopathy: Validation of the boston criteria. Neurology 2001, doi:10.1212/WNL.56.4.537.
- Charidimou, A.; Boulouis, G.; Gurol, M.E.; Ayata, C.; Bacskai, B.J.; Frosch, M.P.; Viswanathan, A.; Greenberg, S.M. Emerging concepts in sporadic cerebral amyloid angiopathy. Brain 2017, 140, 1829–1850, doi:10.1093/brain/awx047.
- Godoy, D.A.; Piñero, G.R.; Koller, P.; Masotti, L.; Napoli, M. Di Steps to consider in the approach and management of critically ill patient with spontaneous intracerebral hemorrhage. World J. Crit. Care Med. 2015, doi:10.5492/wjccm.v4.i3.213.
- Greenberg, S.M.; Vernooij, M.W.; Cordonnier, C.; Viswanathan, A.; Al-Shahi Salman, R.; Warach, S.; Launer, L.J.; Van Buchem, M.A.; Breteler, M.M. Cerebral microbleeds: a guide to detection and interpretation. Lancet Neurol. 2009.
- Charidimou, A.; Martinez-Ramirez, S.; Reijmer, Y.D.; Oliveira-filho, J.; Lauer, A.; Roongpiboonsopit, D.; Frosch, M.; Vashkevich, A.; Ayres, A.; Rosand, J.; et al. Total MRI small vessel disease burden in cerebral amyloid angiopathy: a concept validation imaging-pathological study. JAMA Neurol. 2016.
- Charidimou, A.; Boulouis, G.; Roongpiboonsopit, D.; Xiong, L.; Pasi, M.; Schwab, K.M.; Rosand, J.; Gurol, M.E.; Greenberg, S.M.; Viswanathan, A. Cortical superficial siderosis and recurrent intracerebral hemorrhage risk in cerebral amyloid angiopathy: Large prospective cohort and preliminary meta-analysis. Int. J. Stroke 2019, doi:10.1177/1747493019830065.
- Boulouis, G.; Charidimou, A.; Jessel, M.J.; Xiong, L.; Roongpiboonsopit, D.; Fotiadis, P.; Pasi, M.; Ayres, A.; Merrill, M.E.; Schwab, K.M.; et al. Small vessel disease burden in cerebral amyloid angiopathy without symptomatic hemorrhage. Neurology 2017, doi:10.1212/WNL.0000000000003655.
- Charidimou, A.; Imaizumi, T.; Moulin, S.; Biffi, A.; Samarasekera, N.; Yakushiji, Y.; Peeters, A.; Vandermeeren, Y.; Laloux, P.; Baron, J.C.; et al. Brain hemorrhage recurrence, small vessel disease type, and cerebral microbleeds: A meta-analysis. Neurology 2017, doi:10.1212/WNL.0000000000004259.
Reviewer 2 Report
The authors in this study, compared two groups: one composed of 60 patients with cerebral-amyloid angiopathy and the other 19 controls subject and measured the AQP4 levels. They did not identify difference between patients and controls but identified that patients with more than one lobar hemorrhage or patients with more than 4 microbleeds detected by MRI had lower AQP4 levels compared to controls.
This study has major problems:
- It is not a case control-study, there is a huge difference between both groups
- The controls should have been matched for age, sex and risk factors
- The criteria of inclusion for patients could be discussed : for a preliminary study more strict criteria could have been used : Only probable and certain CAA (with supporting pathology) could have been included.
- The authors used 2 cohorts for patients, what is the rational for this ?
- Time of blood collection is highly variable 12 ±6 for cohort 1 and 15.8 ±17 months for cohort 2.
- Delay of functional outcome measures are highly variable between patients
- What was the primary outcome? If I understood correctly it was the Rankin score but the data on functional outcome only appears in figure 2.
Author Response
The authors in this study, compared two groups: one composed of 60 patients with cerebral-amyloid angiopathy and the other 19 controls subject and measured the AQP4 levels. They did not identify difference between patients and controls but identified that patients with more than one lobar hemorrhage or patients with more than 4 microbleeds detected by MRI had lower AQP4 levels compared to controls.
We want to thank the reviewer for the time taken to review our manuscript. Additionally, the manuscript has been carefully revised by MDPI English Editing Services to improve the language and format of the document.
This study has major problems:
- It is not a case control-study, there is a huge difference between both groups
The reviewer is right with this statement. The original aim of the study was the discovery of new biomarkers related to CAA pathology and the potential association of these markers with the main radiological features of the disease. As part of the global study, we planned to compare the levels of circulating AQP4 between CAA-ICH cases and controls. For this purpose, a sex and age balanced control group was selected from the ISSYS longitudinal cohort (page 3, methods section, highlighted in yellow), considering subjects free of dementia and stroke at the baseline visit and without brain infarction or hemorrhagic lesions in the MRI, as specified in pages 3 and 4 (methods and results sections, highlighted in yellow). Therefore, we do not believe that there is a huge difference between both groups. As it is shown in Table 1 and reported in page 5 (result section, highlighted in yellow) of the manuscript, no differences were observed regarding age, sex or APOE genotype between controls and cases. The main differences were observed in the radiological parameters because controls were expressly selected for not having brain hemorrhages in the MRI to try avoiding the presence of CAA pathology.
- The controls should have been matched for age, sex and risk factors
Controls were matched by age and sex as shown in Table 1, although it was not properly specified in the original version of the manuscript. We apologize for the confusion. Study population description has been modified in the new version of the manuscript in order to clarify this point (page 3). Moreover, the relevant vascular risk factors (hypertension, diabetes, dyslipidemia) of control participants have been included in Table 1 in the new version of the manuscript. All control subjects had hypertension as they were selected from the ISSYS longitudinal cohort as specified in methods section in page 3. Therefore, we were unable to match controls according to all vascular risk factors. However, it is worth mentioning that no differences were observed in the levels of AQP4 between controls and cases. In addition, we also verified that AQP4 levels were not associated with any of these vascular risk factors in the whole cohort.
- The criteria of inclusion for patients could be discussed: for a preliminary study more strict criteria could have been used: Only probable and certain CAA (with supporting pathology) could have been included.
As specified in the Study population section (page 3, methods section, highlighted in yellow), all CAA-ICH patients accomplished a CAA diagnosis according to the modified Boston criteria. According to these criteria, 80% of the patients were classified as probable or probable CAA with supporting pathology, as specified in the first row of Table 2. We agree with the reviewer that the optimal criteria should have been not including patients with a possible CAA diagnosis. However, this study has been part of a multicenter effort to obtain a reasonable sample size to achieve our objective considering the restrictive inclusion criteria to ensure a clear diagnosis of CAA.
Nevertheless, we have re-analyzed the data according to reviewer recommendation and including only the 48 patients diagnosed as probable or probable CAA with supporting pathology. We have included in the present document a brief summary of the re-analysis. As it is shown in the figure, no substantial changes were found in this new re-analysis compared to the results of the original manuscript. First, no significant differences were found in the levels of AQP4 between controls (ng/mL: 2.12 [1.63-2.67]) and cases (ng/mL: 1.88 [1.41-4.27], p=0.867) in the re-analysis. Moreover, circulating AQP4 levels remain significantly lower in patients presenting ≥2 lobar ICHs and patients with ≥5 lobar CMBs (Figure A & B). The negative correlation between the number of lobar ICHs and AQP4 serum levels was also maintained (r= -0.316; p=0.029). Therefore, we consider that the data obtained in our study is conclusive and can be extendible to patients with lobar ICH.
Figure. Association between AQP4 levels and hemorrhagic events in patients diagnosed as probable or probable CAA with supporting pathology. (A) Boxplot distribution according to the number of symptomatic ICHs (1 ICH, n=28; ≥2 ICH, n=20). (B) Boxplot distribution according to the number of lobar CMBs (<5 CMB, n=22; ≥5 CMB, n=26).
- The authors used 2 cohorts for patients, what is the rational for this?
We thank the reviewer for pointing out this relevant point. Reviewer 1 also remarked on this topic and we have attempted to clarify it by modifying Supplemental Figure 1 (page 11). The criteria for dividing patients into two cohorts were the time of functional outcome’s last evaluation, as specified in page 3 (methods, highlighted in yellow). As it is shown in Supplementary Figure 1 (page 11), in subcohort 1, the functional evaluation was undertaken at the follow-up visit, when blood was drawn. However, in the other part of the cohort (subcohort 2), the modified Rankin was evaluated much later than blood collection. In this regard, the information provided in blood samples from each subcohort is not comparable, especially in terms of potential utility of circulating biomarkers for long-term functional prediction. In this regard, in subcohort 1, we cannot consider AQP4 levels in blood as a predictor of outcome because Rankin was evaluated at the same time as when blood was collected. In subcohort 2, as blood samples were collected 34.4 ± 24.8 months before Rankin’s evaluation, we can consider studying whether blood AQP4 levels could be a long-term prognostic biomarker.
For this reason, and although we assume that the ideal situation would have been to obtain functional outcomes at the same time for the whole cohort, the characteristics of our recruitment protocol do not allow us to analyze the AQP4 levels in a unique cohort in terms of functional outcome. We recognize that it is an important limitation in the study, so we already discussed in page 10 (highlighted in yellow).
- Time of blood collection is highly variable 12 ± 6 for cohort 1 and 15.8 ± 17 months for cohort 2.
We fully agree with the reviewer’s comment. The primary outcome of our research line was to find out specific biomarkers of CAA and to detect potential association of these markers with the main radiological hallmarks of the disease. The discovery phase was planned in a chronic stage of the disease, at least 2 months after the last ICH, to avoid confounding factors due to the initial inflammatory process, but without a limit time after the last ICH event. Because the progression of CAA may be understood as a chronic pathological process, we considered that the recruitment time of the ICH-CAA cohort was not a limiting factor to achieve the primary objective. Therefore, blood samples were obtained at the routine clinical follow-up visit after the ICH, which could explain this high variability.
However, in the present study, as specified in the first row of Table 4, it is important to note that no correlation was found between AQP4 levels and the interval between the last ICH and the date of blood collection. We agree with the reviewer that this point might be considered as an important limitation of the study, as we already stated in page 10 in the discussion section (highlighted in yellow).
- Delay of functional outcome measures are highly variable between patients
The reviewer is right with this statement. However, functional outcome was not the primary aim of the study. In fact, we analyzed the association of circulating AQP4 levels with other clinical variables in the cohort, including functional outcome evaluation, as a secondary outcome. Because the time of the Rankin evaluation was not the same within the total CAA-ICH cohort, it was divided into two subcohorts as shown in Supplementary Figure 1. It is important to note that no correlation was found between AQP4 levels and the interval between the last ICH and Rankin evaluation (r=0.078; p=0.558). Additionally, we have modified Supplemental Figure 1 to simplify and clarify the time intervals in both cohorts. We also consider that the high variability in the time of the functional evaluation is an important limitation of the study, as we remarked in page 10 in the discussion section (highlighted in yellow).
- What was the primary outcome? If I understood correctly it was the Rankin score but the data on functional outcome only appears in figure 2.
We acknowledge this reviewer’s comment and apologize for the misunderstanding. Rankin score was not the primary outcome. The primary outcome of the study was to study whether circulating AQP4 could be an indicative biomarker for CAA pathology and to analyze the potential association of this protein with the main radiological hallmarks of the disease. A secondary outcome of the study included the analysis of potential associations of circulating AQP4 levels with functional variables, such as cognitive impairment or Rankin score. In order to clarify the aims of the study, we have modified the last paragraph of the introduction section (page 2).
Round 2
Reviewer 1 Report
The manuscript is much improved
I would suggest to include as reference Gatti et al. 2020
Author Response
Thank you for considering our article for publication in the Journal of Clinical Medicine. We truly appreciate all the constructive comments and suggestions from both reviewers which have been very helpful in improving the manuscript.
We appreciate the reviewer’s 1 suggestion and we have included the reference by Gatti et al., 2020 in the revised manuscript (introduction section, reference number (#17). This is a very inetresting review that provides a global vision of the disease, as well as current challenges for the design of future research studies in the CAA field.
We hope that with this minor change, the reviewer and the editor will find our manuscript more appealing and suitable for publication in the Journal of Clinical Medicine.
Reviewer 2 Report
The manuscripti is now acceptable.
Author Response
Thank you for considering our article for publication in the Journal of Clinical Medicine. We truly appreciate all the constructive comments and suggestions from both reviewers which have been very helpful in improving the manuscript.